# Omega-3 Fatty Acids for the Management of Osteoarthritis: A Narrative Review

**DOI:** 10.3390/nu14163362

**Published:** 2022-08-16

**Authors:** Dean M. Cordingley, Stephen M. Cornish

**Affiliations:** 1Applied Health Sciences, University of Manitoba, Winnipeg, MB R3T 2N2, Canada; 2Pan Am Clinic Foundation, 75 Poseidon Bay, Winnipeg, MB R3M 3E4, Canada; 3Faculty of Kinesiology and Recreation Management, University of Manitoba, Winnipeg, MB R3T 2N2, Canada; 4Centre for Aging, University of Manitoba, Winnipeg, MB R3T 2N2, Canada

**Keywords:** osteoarthritis, fish oil, inflammation

## Abstract

Osteoarthritis (OA) is a disease which results in degeneration of cartilage within joints and affects approximately 13.6% of adults over 20 years of age in Canada and the United States of America. OA is characterized by a state of low-grade inflammation which leads to a greater state of cellular catabolism disrupting the homeostasis of cartilage synthesis and degradation. Omega-3 polyunsaturated fatty acids (PUFAs) have been postulated as a potential therapeutic treatment option for individuals with OA. Omega-3 PUFAs are recognized for their anti-inflammatory properties, which could be beneficial in the context of OA to moderate pro-inflammatory markers and cartilage loss. The purpose of this narrative review is to outline recent pre-clinical and clinical evidence for the use of omega-3 in the management of OA.

## 1. Introduction

Osteoarthritis (OA) is a disease that leads to the degeneration and loss of cartilage within joints, which is initiated by micro and macro-injury and results in a pro-inflammatory response [1]. It is this molecular imbalance which results in anatomical and physiological changes (i.e., cartilage degradation, loss of joint function, and a pro-inflammatory state within the joint) [1]. In Canada, 13.6% of individuals over the age of 20 years have been diagnosed with OA, with prevalence increasing with age [2]. In the United States of America, OA is estimated to affect 32.5 million adults [3], which is approximately 13.6% of the adult population [4]. On a global scale, it is estimated that there are 528 million people with OA, accounting for 31% of all people living with a musculoskeletal condition (~1.71 billion) [5].

Our understanding of OA pathophysiology has progressed but many gaps remain (for review of the pathophysiology of OA, see [6]). OA can be classified as primary OA when it arises spontaneously and not as a result of a traumatic incident, or secondary when it arises following a trauma or due to biomechanical misalignment [7]. Therefore, OA does not stem from a singular influence, but rather can result from multiple factors including inflammatory, biochemical, metabolic, trauma, and biomechanical [8,9]. Numerous pro-inflammatory cytokines, including interleukin-1β (IL-1β), IL-6, IL-8, and tumor necrosis factor alpha (TNF-α), and signaling pathways (nuclear factor kappa light chain enhancer of activated B cells (NF-κB), c-Jun N-terminal kinases (JNK), protein kinase B (Akt), mitogen-activated protein kinase (MAPK), signal transducer and activator of transcription 6 (STAT6), wingless and Int-1/beta-catenin (Wnt/β-catenin), and mammalian target of rapamycin (mTOR)) are implicated in the development of OA [10]. These cytokines and signaling pathways could enhance chondrocyte catabolism and inhibit anabolism, resulting in a disruption of the homeostasis of the cartilage matrix [8] and cartilage degradation [11].

Numerous pharmacological treatments have been evaluated to improve pain associated with OA (for a review, see [12]) but have had limited success [13,14,15]. Non-steroidal anti-inflammatory drugs (NSAIDs; oral and topical) are often used to treat pain since they decrease prostaglandins by suppressing the enzyme activity of cyclooxygenase [12]. However, NSAIDs only have a small-to-moderate effect on pain reduction in OA [16,17] and frequently result in side-effects [18]. Acetaminophen is another commonly used analgesic to manage OA pain; however, it is unclear if its use increases the risk of adverse events and only provides minimal benefits for pain and function [19]. Opioids are another pharmacological option; however, their use is not recommended over nonopioid options due to the potential risks associated with their consumption [20]. There are also some emerging disease-modifying drugs which target the management of cartilage integrity rather than symptoms such as NSAIDs, acetaminophen, and opioids [12]. However, regulatory bodies have not approved any effective disease-modifying therapies [12]. With the lack of success of pharmacological treatments, and some evidence of dietary interventions improving rheumatic diseases symptoms (including rheumatoid arthritis), nutrition-based interventions for OA are of interest [21].

As previously mentioned, joint specific inflammation appears to be an important factor in the pathogenesis of OA [22], suggesting that nutritional interventions incorporating nutrients with anti-inflammatory properties could be beneficial. Omega-3 polyunsaturated fatty acids are recognized for their anti-inflammatory properties (for reviews, see [23,24,25,26]). Omega-3 PUFAs are commonly consumed in fatty fish and seafood, cereal products, seeds, nuts, and vegetables [27,28]; however, western diets tend to be high in omega-6 fatty acids [29]. More recently, dietary supplements containing omega-3 PUFAs have become a popular source for consumption which is available at most grocery, drug, and nutritional-supplement stores. When consuming an omega-3 dietary supplement (or any other dietary supplement), it is important to understand the potential adverse effects [30,31]. Although generally determined to be safe and tolerated [31], omega-3 fatty acid supplements may exacerbate bleeding and anticoagulation in individuals consuming anticoagulants [32], exacerbate antidepressant drug effects, cause gastrointestinal discomfort, and possibly lead to dysgeusia [30]. Omega-6 fatty acids result in the formation of arachidonic acid (ARA), a precursor for eicosanoids (such as prostaglandins and 4-series leukotrienes), which stimulate pro-inflammatory cytokine secretion [23]. The omega-3 fatty acids docosahexaenoic acid (DHA) and eicosapentaenoic acid (EPA), however, can mediate some mechanistic pathways of inflammation [24]. The fatty acid composition of cell membranes, which are altered with the consumption of foods or supplements high in omega-3s, appears to be an important factor in the anti-inflammatory effects of omega-3s [24]. The fatty acid concentration of cell membranes is associated with modifications in cell signaling and gene expression, and the genesis of lipid rafts [24,26]. Similar to ARA, EPA results in the formation of eicosanoids; however, they are less potent, and resolvins [24], which alter the inflammatory and catabolic response, decreasing apoptosis induced by oxidative stress in OA chondrocytes [33]. Furthermore, DHA results in the formation of resolvins, as well as protectins and maresins [24]. Therefore, it is possible that supplementing the diet with omega-3 fatty acids could improve OA symptoms.

The purpose of this review is to outline the current in vitro and in vivo evidence supporting and refuting the efficacy of omega-3 fatty acids for managing and treating OA.

## 2. Discussion

### 2.1. In Vitro and Animal Studies

Obesity is a known risk factor for the development of OA. Evidence suggests that the fatty acid composition of synovial and serum fluid can predict OA resulting from obesity [34]. Six-week-old male mice (C57BL/6J) were fed either a low-fat (10% kcal) or a high-fat diet (60% kcal) high in either saturated fatty acid, or omega-6 or omega-3 PUFAs (polyunsaturated fatty acids) for 24 weeks. After 10 weeks of diet consumption, all mice underwent medial meniscal destabilization of the left hind limb to induce OA and an ear punch to evaluate wound healing. Following the full 24-week protocol, mice were sacrificed, and it was determined that serum omega-3 concentration and the omega-3:omega-6 ratio were negatively correlated with OA severity and wound size, while the omega-6 profile was positively correlated with OA severity and synovitis (assessed by evaluating stromal cell density and lining layer thickness). These results suggest that omega-3 PUFAs could be beneficial in the management of OA, while omega-6 PUFAs may be detrimental.

In addition to the work by Wu et al. [34], more evidence supports the suggestion that the ratio of omega-6 and omega-3 polyunsaturated fatty acids may be important in OA pathogenesis [35]. In both cell culture and murine models (Sprague Dawley rats, 8 weeks of age), a low ratio of omega-6:omega-3 PUFAs inhibits matrix metalloproteinase 13 (MMP13) expression. MMP13 causes the degradation of connective tissue and is important in OA progression and pathology. Human chondrocytes were cultured in varying ratios (1:1 to 10:1) of omega-6 (linoleic acid) and omega-3 (α-linolenic acid). Low levels of omega-6 to omega-3 ratio decreased MMP13 mRNA expression and protein concentrations. In the murine model, rats were injected subcutaneously in the right hind paw with Freund’s complete adjuvant to model arthritis and fed diets ranging in omega-6 to omega-3 ratio of 1:1 to 8:1 for 6 weeks prior to cervical spine dislocation. Low ratios of omega-6 to omega-3 (i.e., 1:1 and 2:1) resulted in decreased MMP13 and interleukin-1 concentrations (both evaluated via enzyme-linked immunosorbent assay (ELISA)) as well as a decreased rate of paw swelling. Additionally, the low ratio of PUFAs acted to protect cartilage damage as a result of the induced arthritis.

Animal and in vitro models have demonstrated that omega-3 fatty acid docosahexaenoic acid (DHA) has the ability to protect cartilage following anterior cruciate ligament (ACL) transection in rats [36]. Male Sprague Dawley rats (12 weeks of age) underwent a sham surgery, ACL transection with placebo tail injection (1 mg/kg of vehicle every second day), or ACL transection with DHA tail injection (1 mg/kg DHA every second day). DHA treatment resulted in decreased bone loss and angiogenesis (determined by micro-CT of the tibial osteochondral unit) compared to the group which received the placebo. The authors suggest that DHA protects against the degradation of cartilage following ACL transection in rats. Additionally, the authors established that DHA may protect against the progression of OA. In cell cultures, the administration of DHA inhibited receptor activator of nuclear factor kappa-B ligand (RANKL) expression, which mediates osteoclast formation and bone remodeling. The expression of RANKL is high in bone and joint diseases such as OA, and therefore overexpression may be detrimental and increase osteoclast formation, which would reduce bone [36].

Utilizing a transgenic mouse model, Kimmerling et al. [37] support the idea that the circulating fatty acid profile is a risk-factor for OA associated with obesity. In the transgenic model, omega-6 PUFAs dehydrogenate to omega-3 PUFAs as a result of omega-3 desaturase, which is encoded by the *fat-1* gene [38]. Therefore, transgenic *fat-1* and wild-type (WT) littermates can consume the same diet but result in different omega-3 lipid profiles. At 5 weeks of age, *fat-1* and WT littermates were randomized to a control diet or a diet high in omega-6 followed by surgery at 16 weeks of age to destabilize the medial meniscus of the left hind limb to induce post-traumatic OA. All mice were sacrificed at 28 weeks of age. At sacrifice, this model resulted in *fat-1* mice having a lower omega-6:omega-3 ratio compared to their WT littermates, who consumed the same high-fat diet with no difference in body weight. Compared to the WT mice, *fat-1* mice had lower OA severity and synovitis.

The omega-3 fatty acid DHA is an agonist of G-protein coupled receptor 120 (GPR120), which has a primary role in regulating free fatty acid and metabolism homeostasis and inhibits inflammation [39]. GPR120 may be an important mediator of inflammation during the development of OA with DHA as a possible mechanism of activation [40]. Chen et al. [40] utilized murine and cell culture models to investigate the role of GPR120 and its agonist DHA in the development of OA. First, WT and GPR120 knockout (KO) mice underwent anterior cruciate ligament transection (ACLT) to induce OA. It was found that GPR120 KO mice showed faster OA development and greater inflammation and cartilage degradation following ACLT than the WT mice. Second, cartilage samples were collected from individuals with OA undergoing total knee arthroplasty (age = 62.3 ± 4.5 years) and control subjects with no history of OA undergoing surgery for a bone fracture (age = 58.8 ± 3.6 years). To induce inflammation, cells were treated with 50 ng/mL of tumor necrosis factor alpha (TNF-α) only or TNF-α and DHA. Overall, GPR120 was downregulated in tissue samples from the OA patients compared to the control subjects, and expression of proinflammatory genes was reduced compared to the group which received TNF-α only. This study identifies the important role of GPR120 in OA progression and the possible benefit of DHA as an agonist.

Derivatives of omega-3 fatty acids may also improve OA pathogenesis [33]. Resolvins are one type of derivative formed from omega-3 fatty acids, where E-series resolvins (RvE) are derived from EPA and D-series resolvins (RvD) are derived from DHA [41], and may be effective in resolving inflammation [42]. Benabdoune et al. [33] obtained chondrocytes from humans with OA undergoing total knee arthroplasty which were then treated with RvD1 either prior to or following treatment with 1 ng/mL of interleukin-1β (IL-1β). It was found that RvD1 can inhibit inflammatory and catabolic mediators associated with OA and prevent apoptosis in OA chondrocytes. These data suggest that omega-3 fatty acid derivative RvD1 may improve OA pathogenesis. The benefits of RvD1 have been corroborated in an obese mouse model where OA was induced by meniscal destabilization [43]. RvD1 treatment can decrease synovial pro-inflammatory markers and can decrease cartilage degradation in response to high-fat diet-induced OA.

Maresin-1 is metabolized from DHA and is postulated to have anti-inflammatory effects [44]. Lu et al. [45] identified whether treatment with endogenous maresin-1 provided a therapeutic effect in rats with induced OA. Thirty 8-week-old Sprague Dawley rats were randomized to a control group (intra-articular knee injection with 50 µL sterile saline), a group with OA (induced with intra-articular knee injection of 1 µg of monosodium iodoacetate and 50 µL sterile saline), or a group with OA and treated with maresin-1 (intra-articular knee injection of 1 µg of monosodium iodoacetate and 50 µL sterile saline, and another injection with 10 ng maresin-1 and 50 µL sterile saline). Injections were given in both knees twice per week for 4 weeks. The injection of 1 µg of monosodium iodoacetate resulted substantial tibial femoral cartilage degradation; however, the maresin-1 injection appeared to be protective and resulted in less cartilage damage. The induction of OA resulted in decreased type II collagen in the cartilage and increased MMP13 in synovium. However, the group which was also treated with maresin-1 had greater type II collagen content in the cartilage and lower MMP13 concentrations in synovium compared to the OA group, which did not receive maresin-1 treatment.

The results of pre-clinical investigations show that treatment with omega-3 and its derivatives/metabolites could be therapeutic and a possible treatment option in the management of OA. Evidence suggests that omega-3 PUFAs may protect against cartilage loss in OA and can decrease inflammatory markers within the joint synovial fluid. However, the evidence in human clinical populations is much less concise.

### 2.2. Human Studies

The first study investigating omega-3 PUFAs in humans with OA was in 1989 [46]. The study included 26 adults aged 52–85 years (female, *n* = 21; men, *n* = 5) diagnosed with OA and were still symptomatic following 2 weeks of treatment with ibuprofen. Participants were randomized to receive either a placebo or an EPA oil (10 mL EPA) for 6 months along with their ibuprofen and were assessed for pain and activities of daily living (100 mm visual analogue scale (VAS)) during the last week of each month. Although statistical differences were not found, based on the difference in improvement for both pain (Placebo= 9.2-point improvement vs. EPA = 15.7-point improvement) and interference in activities of daily living (Placebo = 4.1-point improvement vs. EPA = 10.2-point improvement) by week 24, the authors suggest that it would benefit from further follow-up. However, follow-up research by the same group did not support the notion that fish oil containing omega-3 fatty acids was beneficial in OA management [47]. A total of 86 participants (age range = 49–87 years) diagnosed with OA by their general practitioner (88% confirmed with radiological imaging) were randomized to receive either a cod liver oil supplement (10 mL cod liver oil containing 786 mg EPA) or placebo (10 mL olive oil) to consume along with their current non-steroidal anti-inflammatory drug (NSAID) for 24 weeks. Treatment with cod liver oil did not result in a difference in reported pain or disability (assessed on a 10 cm visual analogue scale) compared to the group which received olive oil. This study suggests that cod liver oil does not result in additional benefits for OA treatment when incorporated with NSAID. However, only the EPA content of the cod liver oil is reported when DHA may be a more important omega-3 for reducing pro-inflammatory cytokines [48]. The cod liver supplement utilized may have had an insufficient DHA to EPA ratio. Furthermore, olive oil has known anti-inflammatory and antioxidant properties, which may not be the most appropriate choice of control oil in studies of this nature [49].

Another study found that an omega-3 supplement resulted in no greater improvement in Western Ontario and McMaster Universities Arthritis Index (WOMAC) score compared to glucosamine sulfate alone in participants with moderate-to-severe hip or knee OA [50]. A total of 177 participants (age = 62.3 ± 7.8 years) were randomized to consume three capsules per day of glucosamine sulfate alone (each capsule contained 500 mg glucosamine sulfate, 444 mg mixed oil without EPA or DHA (70% palm oil, 15% rapeseed oil, 15% sunflower oil), 120 µg vitamin A, 0.75 µg vitamin D, and 1.5 mg vitamin E) or combined with an omega-3 supplement (each capsule contained 500 mg glucosamine sulfate, 444 mg fish oil, 200 mg omega-3 fatty-acids, 120 µg vitamin A, 0.75 µg vitamin D, and 1.5 mg vitamin E) for 26 weeks. There was no difference in the number of participants from each group which met the target decrease in the WOMAC pain score of > 20% (omega-3 = 92.2% vs. glucosamine sulfate alone = 94.3%), but when the threshold was increased to a WOMAC pain score of > 80% results favored the omega-3 group (omega-3 = 44% vs. glucosamine sulfate alone = 32%; *p* = 0.044). This suggests that glucosamine sulfate improves OA pain, and the inclusion of omega-3 may increase the number of individuals who experience a greater reduction (>80%) in pain. The WOMAC comprises a set of questionnaires to evaluate pain, stiffness, and functional limitations in individuals with OA.

There is evidence supporting the use of omega-3 fatty acids in the management of OA. A study investigated whether Green Lipped Mussel (GLM) extract (600 mg per day for 12 weeks) could improve pain and quality of life in 80 patients (age = 66.4 ± 10 yrs) with hip or knee OA (moderate to severe OA) [51]. It was determined that GLM did not result in differences in pain (WOMAC pain scale and 100 mm visual analogue pain scale) or quality of life (OA quality of life score) compared to placebo (600 mg corn oil). However, the group which consumed GLM had improved joint stiffness. It was also found that, 3 weeks following the treatment period, participants who consumed GLM were using less analgesic (paracetamol) than the group which received placebo. However, another study which utilized GLM as the source of omega-3 PUFAs identified benefits for pain management [52]. In total, 80 participants between the age of 46 and 80 with knee OA (GLM, age = 62.1; placebo, age = 62.9) were randomized to receive either GLM (four capsules of Lyprinol^®^ per day for 2 months followed by two capsules per day for the remaining 4 months) or placebo (olive oil at the same number of capsules and duration as the GLM group). The dose of omega-3 and other contents of the treatment is not noted, and participants were able to use acetaminophen for pain management. When results were adjusted to account for acetaminophen use, participants receiving GLM reported a greater decrease in pain (assessed with a 100 mm VAS) at weeks 8, 12, and 24, and a greater improvement in the patient’s global assessment of arthritis (5-point Likert scale) at weeks 12 and 18 compared to placebo. There was no difference in the physician’s global assessment of arthritis (5-point Likert scale), the Chinese Oxford Knee Score (questionnaire to assess pain and function of the knee), or Chinese Arthritis Impact Measurement Scales 2—short form (questionnaire to assess physical aspects, upper limb function, self-care, social activities, and psychology). A study by Jacquet et al. [53] randomized 81 participants (mean age = 57.5 years, range = 28–84 years) with knee or hip OA to consume a commercial food supplement (contained fish oil with omega-3 and omega-6, *Urtica dioica*, zinc, and vitamin E) or placebo for 3 months. It was found that the group which consumed the commercial food supplement used less analgesics and NSAIDs, and had better pain, stiffness and function scores determined by WOMAC compared to placebo. However, the dose of each component in the treatment group and the composition of the placebo group are not reported in the manuscript, which does not allow for critical analysis of which component may be responsible for the improvements.

Identifying the optimal dose of omega-3 supplementation for the management of OA would be beneficial. A comparison of high and low-dose omega-3 fatty acids has been completed which produced some unexpected results and requires follow-up investigation [54]. A total of 202 participants (age = 61 ± 10 years) were randomized to consume 15 mL fish oil per day for two years at either a high (18% EPA and 12% DHA providing 4.5 g EPA + DHA) or low-dose (low-dose fish oil and high oleic oil at a ratio of 1:9, resulting in 0.45 g EPA + DHA). Both groups showed improvements in pain and function over time; however, the low-dose treatment resulted in greater improvements in pain and function (determined with WOMAC) compared to the high-dose group. Further investigation into these results and determination of the optimal omega-3 fatty acid dose would improve future research on its efficacy for the management of OA symptoms as this evidence does not indicate that more is necessarily better. However, as previously stated, oleic acid (found to a high degree in olive oil) has anti-inflammatory and antioxidant properties associated with it. Thus, it may not be the optimal oil for comparing and analyzing the anti-inflammatory profile of omega-3 fatty acids [28].

Most recently, Stonehouse et al. [55] evaluated the effects of krill oil (4 g/d containing 0.6 g/d EPA, 0.28 g/d DHA, and 0.45 g/d astaxanthin) on pain, stiffness, physical function, NSAID use, and inflammatory markers in adults (*n* = 235; age = 55.9 ± 6.8 yrs old) diagnosed with mild-to-moderate knee OA (but otherwise healthy). Compared to the placebo condition (mixed vegetable oil), krill oil resulted in greater improvements in pain (krill oil = 17.8% improvement vs. placebo = 12.6% improvement), stiffness (krill oil = 19.5% improvement vs. placebo = 13.1% improvement), and physical function (krill oil = 14.8% improvement vs. placebo = 10.1% improvement), but no differences were found for NSAID use or inflammatory markers. These results indicate that krill oil may improve pain, stiffness and physical function for individuals with mild-to-moderate knee OA, but the true magnitude of improvement is difficult to identify as the placebo condition resulted in significant improvements from baseline as well.

Polyunsaturated fatty acids may also aid in the prevention of radiographic progression of OA [56]. A prospective study of 2092 individuals (age, 62.4 ± 9.0 years) with knee OA were followed for up to 4 years with yearly radiographic imaging, to determine joint-space width and monitoring of dietary fat intake (Block Brief Food Frequency Questionnaire). It was determined that participants who consumed more monounsaturated and PUFAs experienced less joint space loss over the 4 years. This preliminary evidence suggests that omega-3 fatty acids (which are polyunsaturated in structure) may delay the anatomical changes of a knee joint with OA. Currently, according to clinicatrials.gov (accessed 8 August 2022), ongoing clinical trials investigating omega-3 PUFAs on OA symptoms and outcomes include one clinical trial completed but without results (NCT04121533) and one with an unknown recruitment status (NCT02333084).

The current state of the evidence for the use of omega-3 PUFAs as a therapeutic in human OA patients is mixed. Limitations to the current literature include differences in omega-3 doses, as well as EPA and DHA concentrations, and the use of placebos (i.e., olive oil), which may not be appropriate as they could have their own effects in the OA population. These limitations could contribute to variations in results. More research in this area is required to determine if the pre-clinical data can be supported with clinical data or if the clinical data will not support the use of omega-3 fatty acid supplementation strategies in OA populations. See Table 1 for a summary of all human studies.

## 3. Conclusions

Pre-clinical evidence provides credible proof of human trials in the use of omega-3 PUFAs as a possible treatment option in OA, while human evidence shows that omega-3s may be efficacious but requires further research to determine optimal treatment protocols. It is postulated that the anti-inflammatory properties of omega-3 PUFAs could alleviate the low-grade inflammatory environment associated with OA and slow cartilage catabolism and OA progression, but this requires further exploration. Decreased inflammation and cartilage catabolism may correspond to increased function and improved symptoms associated with OA such as joint pain and stiffness. OA accounts for a large burden on the healthcare system, with knee OA in the United States of America costing UDS 5.7 billion to USD 15 billion annually [57]. If future research concludes that omega-3 treatment regimens (along with additional potentially beneficial nutritional treatments [21]) can improve patient reported outcomes, this could be a cost-effective treatment option resulting in decreased healthcare costs.

## Figures and Tables

**Table 1 nutrients-14-03362-t001:** Omega-3 in humans with osteoarthritis.

Author	Design	Sample	Intervention	Main Results	Conclusions
Stammers et al. (1989) [46]	RCT	N = 26; female, *n* = 21; age range = 52–85 yrs	EPA oil (10 mL/d EPA) and ibuprofen (1200 mg/d) or placebo (oil of undescribed content) and ibuprofen for 6 months	No differences were identified for pain (Placebo = 9.2 point improvement vs. EPA = 15.7 point improvement) or activities of daily living (Placebo = 4.1 point improvement vs. EPA = 10.2 point improvement) utilizing a 100 mm VAS.	Although no statistical differences were observed, as the first study investigating omega-3 PUFAs in patients with OA the authors suggest the results indicate future research would be of value.
Stammers et al. (1992) [47]	RCT	N = 86; female, *n* = 60; age range = 49–87 years	Cod liver oil (10 mL cod liver oil containing 786 mg EPA) and current NSAIDs or placebo (10 mL olive oil) and current NSAIDs for 24 weeks	Cod liver oil did not result in improved pain (cod liver = 1 point worsening vs. placebo = 3 point improvement) or disability (cod liver = 2 point improvement vs. placebo = 4 point improvement) at 6 months as assessed with a 10-cm VAS.	Cod liver oil supplementation does not result in benefits for OA related pain or disability when consumed in addition to the patients regular NSAIDs.
Gruenwald et al. (2009) [50]	RCT	N = 177; female, *n* = 113; age = 62.3 ± 7.8 yrs	3 capsules/d glucosamine sulfate alone (each capsule contained 500 mg glucosamine sulfate, 444 mg mixed oil without EPA or DHA (70% palm oil, 15% rapeseed oil, 15% sunflower oil), 120 µg vitamin A, 0.75 µg vitamin D, and 1.5 mg vitamin E) or combined with an omega-3 supplement (each capsule contained 500 mg glucosamine sulfate, 444 mg fish oil, 200 mg omega-3 fatty-acids, 120 µg vitamin A, 0.75 µg vitamin D, and 1.5 mg vitamin E) for 26 weeks	No difference in number of participants which achieved the target threshold of ≥ 20% decrease in pain score (omega-3 and glucosamine = 92.2% vs. glucosamine alone = 94.3%) but the addition of omega-3s appeared to increase the number of participants reaching the ≥ 80% reduction in pain threshold (omega-3 and glucosamine = 44% vs. glucosamine alone = 32%; *p* = 0.044) evaluated with the WOMAC	The addition of omega-3 PUFAs to a glucosamine sulfate supplement do not improve the number of patients with a decrease in pain meeting the initial study cut off of ≥ 20%, but was beneficial when the cut-off was increased to ≥ 80%.
Stebbings et al. (2017) [51]	RCT	N = 80; female, *n* = 44; age = 66.4 ± 10 yrs	Green-lipped mussel (GLM) extract (600 mg/d) or placebo (600 mg/d corn oil) for 12 weeks	No difference was observed between groups at 12 weeks for pain (evaluated with WOMAC pain scale and 100 mm VAS) or quality of life (evaluated with OA quality of life score). However, GLM resulted in improved stiffness (GLM, median = 3.0 vs. placebo, median = 3.7; *p* = 0.046) and decreased acetaminophen use following the 12-week intervention phase (*p* = 0.001)	In patients with moderate-to-severe OA, GLM did not improve pain, but may be beneficial for stiffness and to decrease acetaminophen use following treatment
Lau et al. (2004) [52]	RCT	N = 80; female, *n* = 69; age, GLM = 62.1 and placebo = 62.9 yrs	GLM (4 capsules of Lyprinol^®^ per day for 2 months followed by 2 capsules per day for the remaining 4 months) or placebo (olive oil at the same number of capsules) for 4 months. The dose of omega-3 and other contents of the treatment is not available in the study	GLM supplementation resulted in a greater decrease in pain (assessed with a 100 mm VAS) at weeks 8, 12, and 24 (GLM = 54.0 ± 15.2 vs. placebo = 67.1 ± 5.5), and a greater improvement in the patients’ global assessment of arthritis at weeks 12 and 18 (GLM = 3.0 ± 0.8 vs. placebo = 3.1 ± 0.7) compared to placebo when results are adjusted for acetaminophen use. There was no difference in the physician’s global assessment of arthritis, the Chinese Oxford Knee Score, or Chinese Arthritis Impact Measurement Scales 2—short form	GLM may result in decreased perceived pain and global assessment of arthritis, but not other assessments of quality of life and physical functioning
Jacquet et al. (2009) [53]	RCT	N = 81; female, *n* = 55; mean age = 57.5; age range = 28–84 yrs)	Commercial food supplement (containing fish oil with omega-3 and omega-6, *Urtica dioica*, zinc, and vitamin E; dose of each compound not reported) or placebo (contents not reported) for 3 months	The commercial food supplement resulted in better pain (supplement = 86.5 vs. placebo = 235.3; *p* < 0.001), stiffness (supplement = 41.4 vs. placebo = 96.3; *p* < 0.001), and function (supplement = 301.6 vs. placebo = 746.5; *p* < 0.001) (evaluated with the WOMAC) and less NSAID consumption compared to placebo (*p* = 0.02)	The commercial food supplement improved pain, stiffness, and function, as well as decreased NSAID use in individuals with OA. However, the lack of information regarding dose and placebo content makes interpretation of results difficult as to which ingredient may result in the observed benefits
Hill et al. (2006) [54]	RCT	N = 202; female, *n* = 100; age = 61 ± 10 years	15 mL fish oil/day at either a high (18% EPA and 12% DHA providing 4.5 g EPA + DHA) or low-dose (low-dose fish oil and high oleic oil at a ratio of 1:9, resulting in 0.45 g EPA + DHA) for 2 years.	Both high and low-dose omega-3 supplement groups improved pain and function (assessed with the WOMAC), but the low-dose group should have greater improvements (pain, mean difference at 24 months = 4.1; *p* = 0.001; and function, mean difference at 24 months = 11.6; *p* = 0.002). Neither group differed in cartilage volume and bone marrow lesion area over the 2 years	Omega-3 PUFAs can improve pain and function, but a low dose may be more beneficial. Future studies investigating the optimal dose for OA patients are required
Stonehouse et al. (2022) [55]	RCT	N = 235; female, *n* = 129; age = 55.9 ± 6.8 yrs	Krill oil (4 g/d containing 0.6 g/d EPA, 0.28 g/d DHA and 0.45 g/d astaxanthin) or placebo (4 g/d mixed vegetable oil containing olive oil, corn oil, palm oil and medium chain triglycerides) for 6-months	Krill oil resulted in greater improvements in pain (krill oil = 17.8% improvement vs. placebo = 12.6% improvement), stiffness (krill oil = 19.5% improvement vs. placebo = 13.1% improvement), and physical function (krill oil = 14.8% improvement vs. placebo = 10.1% improvement) compared to placebo. No changes in NSAID use or inflammatory markers were found.	Krill oil may be beneficial to improve pain, stiffness, and physical function, but not NSAID use or decrease markers of inflammation in individuals with mild-to-moderate knee OA
Lu et al. (2017) [56]	Prospective cohort study	N = 2092; female, *n* = 1230; age, 62.4 ± 9.0 years	N/A- Dietary consumption of MUFAs and PUFAs was monitored with the Block Brief Food Frequency Questionnaire for 4-years.	Increasing quartiles of PUFA consumption was associated with a decreased hazard ratios for joint-space loss (Q2, HR = 1.01 (0.79–1.30); Q3, HR = 0.67 (0.51–0.89); Q4, HR = 0.70 (0.53–0.93)) compared to the bottom quartile (Q1). The top quartile (Q4) of MUFA consumption had a 25% decreased risk of OA progression compared to Q1	Individuals with OA who consumed more MUFAs and PUFAs experienced less joint-space loss within the knee joint

EPA-eicosapentaenoic acid; DHA-docosahexaenoic acid; GLM-green-lipped mussel; MUFAs-monounsaturated fatty acids; NSAID-nonsteroidal anti-inflammatory drugs; OA-osteoarthritis; PUFAs-polyunsaturated fatty acids; VAS-visual analogue scale; WOMAC-Western Ontario and McMaster Universities Arthritis Index.

## Data Availability

Not applicable.

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
