# Peer review of "Omega-3 Fatty Acids for the Management of Osteoarthritis: A Narrative Review"

_nutrients, 2022, doi:10.3390/nu14163362_

Round 1

Reviewer 1 Report

The present manuscript presents an interesting overview of omega-3 fatty acids for the management of osteoarthritis. Please find my suggestions and comments below:

-        Introduction:

o   Authors should present the pathophysiological mechanisms involved in OA

o   Worldwide data on OA incidence should be provided

o   It would be better if authors present the principles of OA pharmacotherapy and the role of each drug class used in therapy.

o   Also, authors should provide some issues regarding omega 3 fatty acid:

§  what are their sources for intake or for obtaining dietary supplements (DS)?

§  what is their mechanism of anti-inflammatory activity?

§  what are the differences between omega 3 and omega 6?

§  Even the products containing omega-3 are DS, their use could be limited by adverse effects (AE). Different studies reported such AE.

§  https://academic.oup.com/jn/article/142/3/592S/4630968

§  https://farmaciajournal.com/wp-content/uploads/2019-03-art-24-Morgovan_Ghibu_Olah_537-544.pdf

§  https://www.ncbi.nlm.nih.gov/pmc/articles/PMC2174995/

-        Discussions:

o   This section presents a small number of studies conducted in OA. Other studies must be added.

o   You should summarize the studies described in a table. Their benefits and limitations should be presented.

o   In section 2.2 authors should provide data on registered clinical trials

-        Authors should present details regarding WOMAC and other tests performed for evaluation of OA.

-        What are the limitations of this study?

Author Response

Thank you to the reviewers for their thorough review of our manuscript titled “Omega-3 Fatty Acids for the Management of Osteoarthritis: A Narrative Review” Manuscript ID: nutrients-1854532

Please find our point-by-point response to the reviewers comments below in bold as well as the revised manuscript with all changes highlighted.

Reviewer 1

The present manuscript presents an interesting overview of omega-3 fatty acids for the management of osteoarthritis. Please find my suggestions and comments below:

-        Introduction:

o   Authors should present the pathophysiological mechanisms involved in OA

Thank you for this recommendation.  A paragraph has been added to the introduction which outlines the pathophysiology of OA.

  • Worldwide data on OA incidence should be provided

We have included data from the World Health Organization on the prevalence of OA globally in the Introduction.

On a global scale, it is estimated that there are 528 million people with OA accounting for 31% of all people living with a musculoskeletal condition (~1.71 billion) (World Health Organization 2022).”

  • It would be better if authors present the principles of OA pharmacotherapy and the role of each drug class used in therapy.

Based on your suggestion we have included a new paragraph in the introduction outlining the current pharmacotherapy for OA.

o   Also, authors should provide some issues regarding omega 3 fatty acid:

  • what are their sources for intake or for obtaining dietary supplements (DS)?

Thank you for this suggestion.  We have added in the following sentences to the introduction.  “Omega-3 PUFAs are commonly consumed in fatty fish and seafood, cereal products, seeds, nuts and vegetables (Richter et al. 2017; Cholewski et al. 2018),  however western diets tend to be high in omega-6 fatty acids (Baker et al. 2016).  More recently, dietary supplements containing omega-3 PUFAs have become a popular source for consumption which is available at most grocery, drug and nutritional supplement stores.”

  • what is their mechanism of anti-inflammatory activity?

We have expanded the explanation of the mechanisms in the introduction.

  • what are the differences between omega 3 and omega 6?

Please the newly expanded section paragraph on mechanisms where we now outline the similarities and differences between omega-3 and omega-6 PUFAs.

  • Even the products containing omega-3 are DS, their use could be limited by adverse effects (AE). Different studies reported such AE.

We have added the following to the introduction outlining some of the AEs associated with omega-3s.

“When consuming an omega-3 dietary supplement (or any other dietary supplement) it is important to understand the potential adverse effects (Ronis et al. 2018; Morgovan 2019).  Although generally determined to be safe and tolerated (Ronis et al. 2018) omega-3 fatty acid supplements may exacerbate bleeding and anticoagulation in individuals consuming anticoagulants (Gross et al. 2017), exacerbate antidepressant drug effects, cause gastrointenstinal discomfort and possibly lead to dysgeusia (Morgovan 2019).”

  • https://academic.oup.com/jn/article/142/3/592S/4630968
  • https://farmaciajournal.com/wp-content/uploads/2019-03-art-24-Morgovan_Ghibu_Olah_537-544.pdf
  • https://www.ncbi.nlm.nih.gov/pmc/articles/PMC2174995/

Thank you for the above references.  They have been included in the manuscript.

-        Discussions:

o   This section presents a small number of studies conducted in OA. Other studies must be added.

We have updated the articles in section 2.2 by adding in (Stammers et al. 1989; Lau et al. 2004; Stonehouse et al. 2022).  Additional manuscripts investigating omega-3s in patients were identified but one also included patients with rheumatoid arthritis and cardiovascular disease and did not break down results to delineate the changes between conditions (Deutsch 2007), while we were unable to access one additional paper (Yazdanpanah et al. 2014).

  • You should summarize the studies described in a table. Their benefits and limitations should be presented.

A table has been added to the manuscript.  We hope it helps summarize the discussed studies.

  • In section 2.2 authors should provide data on registered clinical trials

Thank you for this suggestion.  We have now listed trial registration numbers from clinicaltrials.gov for studies investigation omega-3 PUFAs on symptoms and structural progression in OA patients.

-        Authors should present details regarding WOMAC and other tests performed for evaluation of OA.

       A description of the WOMAC has been added to the paragraph it is first mentioned.

-        What are the limitations of this study?

Reviewer 2 Report

I commend author for this exhastive review providing an account of recent pre-clinical and clinical studies evaluating the omega-3 polyunsaturated fatty acids (PUFAs) for treatment of OA.

I have a few suggestions listed below.

1. Please provide the full form of PUFA in the main manuscript at its first occurrence.

2. Please reconsider writing line 42, as currently, it is hard to follow.

3. One of the limitations of the current clinical evidence is the improper use of comparators in clinical studies. Several studies used olive oil as a comparator that may have its own effect. This is mentioned by the authors as well. I suggest the author mention this in the last paragraph (starting from line 224) as well.

4. Please rewrite the below line, as preclinical evidence cant support the use of PUFA, it can only provide credible evidence for trial in human, but not enough evidence to support the clinical use of PUFA in humans.

"Pre-clinical evidence supports the use of omega-3 PUFAs as a possible treatment option in OA," Line 231

5. I think on the basis of the current (uncertain) evidence author can only advocate for future clinical trials. And the PUFA acting as a treatment option (cos-effective) is a story far away. Hence I suggest updating the conclusion accordingly.

Author Response

I commend author for this exhastive review providing an account of recent pre-clinical and clinical studies evaluating the omega-3 polyunsaturated fatty acids (PUFAs) for treatment of OA.

I have a few suggestions listed below.

  1. Please provide the full form of PUFA in the main manuscript at its first occurrence.

Thank you for identifying this omission.  This has been corrected.

  1. Please reconsider writing line 42, as currently, it is hard to follow.

This line has been revised to the following which we hope improves the clarity.

Omega-3 fatty acids however, form eicosanoids which are less inflammatory (or anti-inflammatory) compared to those formed from omega-6 fatty acids (James et al. 2000). This results in competitive inhibition of enzymes involved in the conversion of arachidonic acid to eicosanoids (James et al. 2000; Calder and Grimble 2002)

  1. One of the limitations of the current clinical evidence is the improper use of comparators in clinical studies. Several studies used olive oil as a comparator that may have its own effect. This is mentioned by the authors as well. I suggest the author mention this in the last paragraph (starting from line 224) as well.

Thank you for the suggestion.  We have updated the limitations section to read “Limitations to the current literature include differences in omega-3 doses, as well as EPA and DHA concentrations, and the use of placebos (i.e., olive oil) which may not be appropriate as they could have their own effects in the OA population.  These limitations could contribute to the variations in results.”

  1. Please rewrite the below line, as preclinical evidence cant support the use of PUFA, it can only provide credible evidence for trial in human, but not enough evidence to support the clinical use of PUFA in humans.

"Pre-clinical evidence supports the use of omega-3 PUFAs as a possible treatment option in OA," Line 231

Thank you for this suggestion.  The sentence has been re-written as “Pre-clinical evidence provides credible evidence for human trials in the use of omega-3 PUFAs as a possible treatment option in OA,”

  1. I think on the basis of the current (uncertain) evidence author can only advocate for future clinical trials. And the PUFA acting as a treatment option (cos-effective) is a story far away. Hence I suggest updating the conclusion accordingly.

We agree with this comment.  The conclusion has been updated to advocate for future clinical trials and only speculates on the possible benefits if future evidence supports the use of omega-3 in the treatment of OA.

Round 2

Reviewer 1 Report

The manuscript has been modified and is suitable for publication.